# $^{40}$Ar/$^{39}$Ar Geochronology, Geochemistry and Petrogenesis of the Volcanic Rocks in the Jiangling Basin, China

Chunlian Wang [1], Kai Yan [1,2,*], Xiaocan Yu [1,*], Jiuyi Wang [1], Dianhe Liu [1,3], Lijian Shen [1], Ruiqin Li [1,3,4] and Chao You [5]

1    MNR Key Laboratory of Metallogeny and Mineral Assessment, Institute of Mineral Resources, Chinese Academy of Geological Sciences, Beijing 100037, China
2    Faculty of Earth Science, University of Iceland, Sturlugata 7, Askja, 101 Reykjavík, Iceland
3    School of earth and space Sciences, Peking University, Beijing 100871, China
4    Department of Geological Sciences and Environmental Studies, State University of New York at Binghamton, New York, NY 13902, USA
5    Institute of Geological Survey, China University of Geosciences, Wuhan 430074, China
*    Correspondence: yankai_ytq@sina.com (K.Y.); xiaocany1988@163.com (X.Y.)

**Abstract:** In this study, $^{40}$Ar/$^{39}$Ar geochronology and major and trace element data were presented for Paleogene basaltic rocks from the Jiangling Basin, China. The volcanic rocks erupted at ca. 53.19–60.78 Ma and belonged to the sub alkaline series. These basaltic rocks are generally characterized by enrichment in large-ion lithophile elements (LILEs) and light rare earth elements (LREEs) ((La/Yb)$_{cn}$ = 6.14–11.72) and lack of Eu anomalies (Eu/Eu* = 0.98–1.09), similar to ocean island basalts. The geochemical signatures of these rocks are similar to hotspot-related Paleogene volcanic rocks in the North China Block and late Cenozoic volcanic rocks in Southeast China. The Cenozoic lithospheric mantle, as well as the Mesozoic basalts that are beneath the northern Yangtze Blocks, might be inherited from the Mesozoic lithospheric mantle. The basaltic rocks from the Jiangling Basin in the northern Yangtze Block were generated from the partial melting of EMII (enrichedmantleII)-like lithospheric mantle due to the intracontinental extension.

**Keywords:** Paleogene; volcanic rock; intracontinental extension; Jiangling Basin; geochronology

## 1. Introduction

Intracontinental volcanic fields are generally characterized by prolonged activity over periods of millions of years [1–10]. They can generate large shield volcanoes and lava flow fields [11,12], or primarily consist of monogenetic volcanic vents [13,14].

In addition, the fundamental characteristics of volcanic fields can also provide information on magma generation [15–19], the timing and frequency of eruptions [4,20,21], and the distribution of volcanoes [22]. The volcanic research can also give some information on the relationship of the volcanoes, basins, faults, and rift zones [10,23–27].

During the Cenozoic, volcanic rocks and extensional tectonics within graben basins occurred widely in the Eastern and Central regions of China. To date, most studies have focused on the Eastern coastal areas, including the area from Northeastern China to the South China Sea Basin, and analyzed the origin of these Cenozoic volcanic basaltic rocks and the differences in lithospheric mantle properties in the North and the South China blocks [28–47]. The studies found that the lithospheric mantle of Northeastern China and the North China Block had the features of enriched mantle I (EMI)-type components [34,48] and that the lithospheric mantle of the Southeastern coast and the South China Sea Basin had features of enriched mantle II (EMII)-type components [31,33,35,49,50].

However, in the central region of China, especially on both sides of the Qinling–Dabie orogenic belt, the understanding of the differences in the properties of the lithosphere since the Cenozoic between the South China and the North China blocks remains incomplete

because the Cenozoic volcanic outcrops have so far provided limited information [33,51]. Therefore, we have selected the volcanic rocks from the Jiangling Basin to the south of the Dabie Mountains to perform more systematic chronological and geochemical research. We aim to understand the timing, petrogenesis and geodynamic setting of the basaltic rocks and to discuss the implications of our results in the context of the mantle source region during the Cenozoic.

## 2. Geological Background, Field Observations, and Petrography

During the Cretaceous to Cenozoic, intraplate rifting and magmatism are widely developed in Eastern China [52,53]. They migrated eastward and ceased during the Late Cenozoic. The geodynamic origin has been attributed to the subduction and rollback of the paleo-Pacific Plate and the India-Asia collision [54,55].

The Jiangling Basin is located in the south of the Dabie Mountain (Figure 1), which represents a Cenozoic rift basin with an area of approximately 28,000 km². Because of the Qinling-Dabie orogeny and orogenic extensional tectonics that impacted the basin, the region has experienced a developmental history of continental basins filled with thick terrigenous clastic series from the Middle Triassic. Most of the mafic volcanic rocks are buried in the basin with only small exposures at the surface (Figures 1 and 2) [56]. The volcanic activity in the basin can be divided into two cycles: late Mesozoic volcanics and Paleogene volcanics [56].

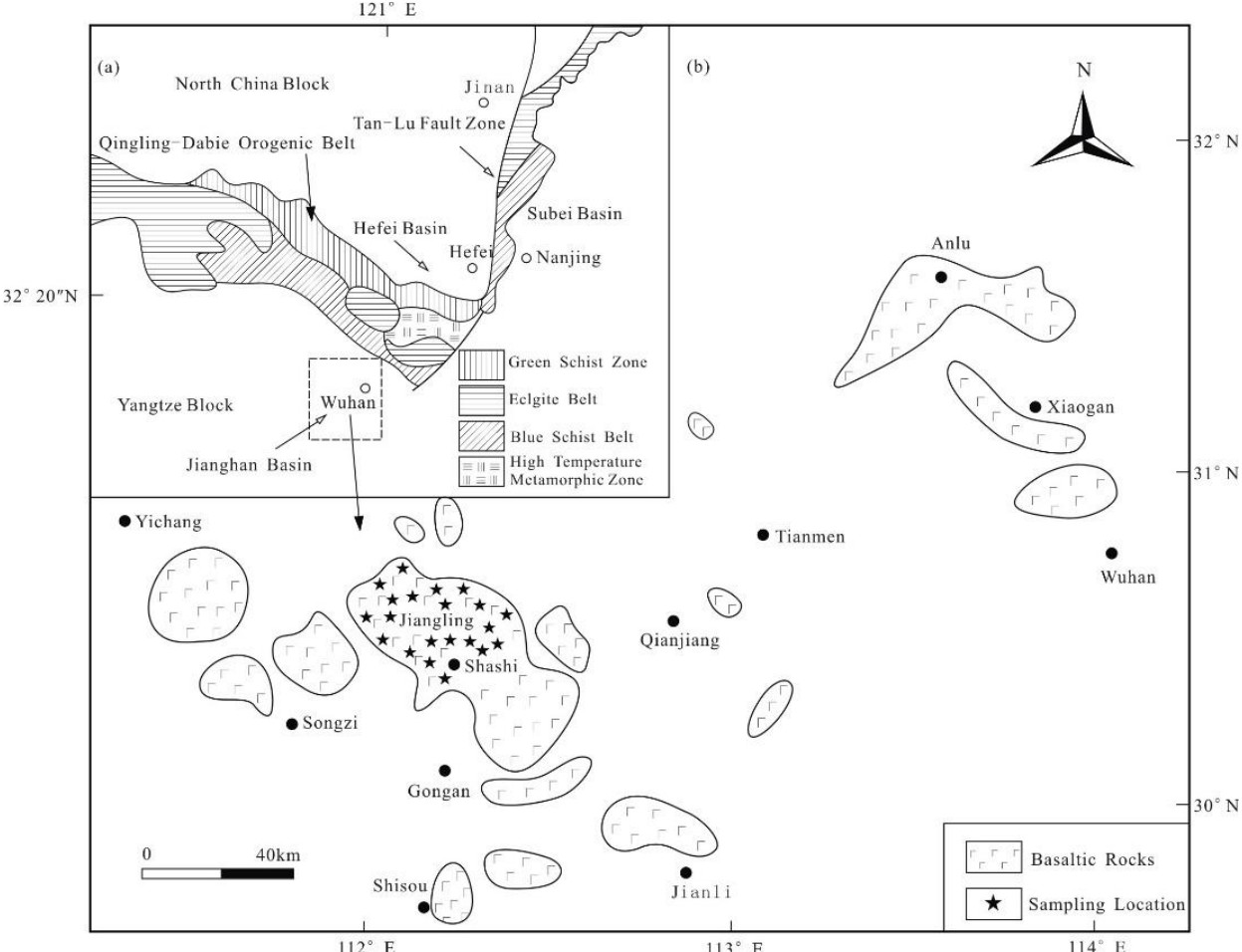

**Figure 1.** (**a**) Tectonic division of Central China. (**b**) Geological map of the Jiangling Basin and sampling locations.

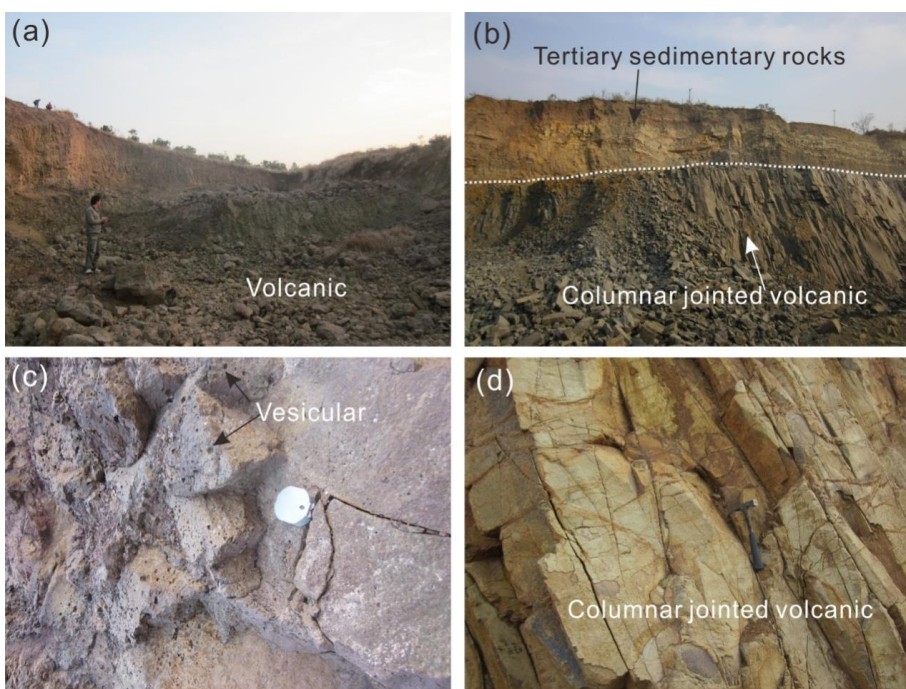

**Figure 2.** Field photographs showing the various lithologies of the volcanic rocks in the Jiangling Basin. (**a**) Fresh volcanic rocks; (**b**) Tertiary sedimentary rocks overlapping volcanic rocks with columnar jointing; (**c**) Vesicular volcanic rocks; and (**d**) Volcanic rocks with columnar jointing.

The Paleogene volcanics are fine-grained and occasionally porphyritic consisting of plagioclase, chlorite, and opaque minerals (Figure 3). Plagioclase occurs as small laths and a few microphenocrysts. The plagioclase laths commonly show quench textures, especially along the peripheries of the pillows. Chlorite is mainly present in the groundmass and formed at the expense of pyroxene. The laths are mostly filled with chlorite and quartz and some are filled with calcite, epidote, and prehnite.

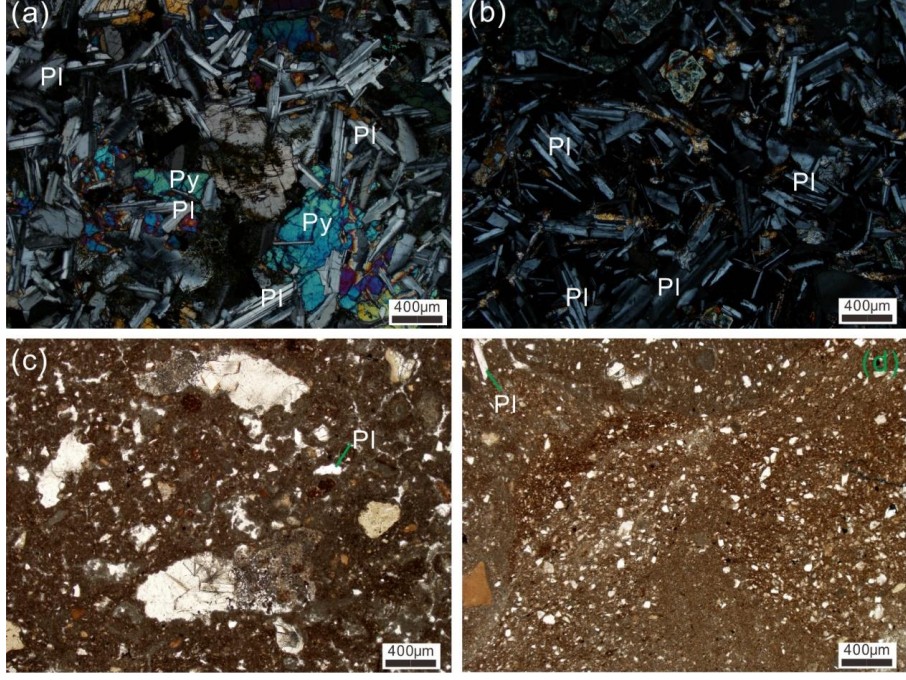

**Figure 3.** Photomicrographs of volcanic rocks from the Jiangling Basin. Py-pyroxene, Pl-plagioclase.

(**a**) intergranular basalt. Olivine undergoes intense alteration, chloritization, and iddingsitization; perpendicular polarized light. Plagioclase is colorless and transparent and has two groups of complete cleavage; long strip, polarization microscope common twin, interference color level gray, negative low protrusion. (**b**) Basalt is mainly composed of plagioclase (Pl) and pyroxene (Py), with weak alteration pairs; perpendicular polarized light. Pyroxene is the most common mineral in ultrabasic rocks and mafic rocks. It is greenish black and has two groups of nearly vertical cleavage, often forming irregular stepped fractures. (**c**) plane polarized light of (**a**). (**d**) plane polarized light of (**b**).

## 3. Sampling and Analytical Methods

We selected the freshest Paleogene volcanic samples (n = 21) for whole-rock geochemical analysis. Each sample was carefully cleaned, crushed, and then ground in an agate mortar to pass through a 200-mesh screen. The major elements were determined on fused glass beads via X-ray fluorescence (XRF) spectrometry. The analyses were performed in the National Research Center for Geoanalysis of Chinese Academy of Geological Sciences. The analytical accuracy was estimated at 1% for $SiO_2$ and 2% for the other oxides. Trace elements, including rare earth elements (REE) and high field strength elements (HFSE), were determined by inductively coupled plasma-mass spectrometry (ICP-MS). The analyses were performed in the National Research Center for Geoanalysis of Chinese Academy of Geological Sciences. Two national standards (GSR3 and GSR5) and three internal standards were measured simultaneously to ensure the consistency of the analytical results. Analytical uncertainties were estimated at approximately 10% and 5% for trace elements with abundances of <10 ppm and >10 ppm, respectively.

Water and $CO_2$ contents were determined by gravimetric techniques in which the sample was heated in a closed container and the water vapor was collected in a separate tube, condensed and then weighed—the detection limit for $H_2O$ and $CO_2$ was 0.01 wt.%.

To precisely constrain the formation age of the volcanic rocks, we performed $^{40}Ar/^{39}Ar$ stepwise laser heating experiments. Selected samples (n = 9) were purified and crushed, repeatedly sieved to uniform mineral grains of 0.5–2 mm, washed in an ultrasonic bath of distilled water for 1 h, and then dried. The argon isotopes were analyzed on a GV Instruments 5400® mass spectrometer with a secondary electron multiplier (Balzers SEV217) under the pulse counting mode and a coherent $CO_2$ 50-W laser at the State Key Laboratory of Isotope Geochemistry, Guangzhou Institute of Geochemistry, Chinese Academy of Sciences. The experimental methods and procedures have previously been described [57,58]. The samples and a monitor standard ZBH-25 biotite with an assumed age of 132.5 Ma [59,60] were irradiated at the 49-2 reactor in Beijing for 48 h. The correction factors for interfering argon isotopes derived from Ca and K were as follows: $(^{39}Ar/^{37}Ar)$ Ca = $8.984 \times 10^{-4}$, $(^{36}Ar/^{37}Ar)$ Ca = $2.673 \times 10^{-4}$ and $(^{40}Ar/^{39}Ar)$ K = $5.97 \times 10^{-3}$. The extraction and purification lines were baked out for 20 h at 150°C with heating tape, and the sample chamber was baked out with a furnace. The static blank of $^{40}Ar$ after 5 min was approximately 2 mV. The experiments began and ended with cold blank analyses, and cold blanks were also measured after every four-step sample analysis. The released gas was purified for 5 to 8 min by two Zr/Al getter pumps operated at room temperature or approximately 400 °C. The purified gas was then analyzed.

Whole-rock Sr, Nd, and Pb isotopic compositions of sixteen samples were analyzed using a Finnigan Triton thermal ionization mass spectrometer at the modern analysis center of the Nanjing University. The detailed procedures were described in Pu et al. (2005). $^{86}Sr/^{88}Sr$ = 0.1194 and $^{146}Nd/^{144}Nd$ = 0.7219 were used for the mass fractionation corrections of Sr and Nd isotopes, respectively. Nd standard JNDi-1 yielded $^{143}Nd/^{144}Nd$ = 0.512117 ± 0.000006 (2σ) and Sr standard NBS-987 gave $^{87}Sr/^{86}Sr$ = 0.710252 ± 0.000007 (2σ). Pb isotopes were given relative to the standard NBS-981 values of $^{206}Pb/^{204}Pb$ = 16.9410, $^{207}Pb/^{204}Pb$ = 15.4944, and $^{208}Pb/^{204}Pb$ = 36.7179, as shown in Collerson et al. (2002) [61].

## 4. Results

### 4.1. Whole-Rock Major Elements

A total of 21 samples of volcanic rocks were analyzed for major elements and trace elements, as shown in Table 1.

The Jiangling volcanic rocks have $SiO_2$ contents of 50.3–54.3 wt.% (Figure 4), which is typical of moderately evolved basalts to andesites. They have moderate contents of $TiO_2$ (1.76–2.0 wt.%) and nearly constant contents of MgO (4.07–6.84 wt.%). Major element variations against MgO are shown in Figure 5. The $Al_2O_3$, $SiO_2$, and $Na_2O$ contents of most samples increase with the decreasing MgO values, which is consistent with low-pressure fractional crystallization of clinopyroxene and plagioclase. Nevertheless, the limited variation in MgO, Cr (162–218 ppm) and Ni (96.4–144 ppm) suggests limited magma differentiation.

In the $Na_2O + K_2O$ vs. $SiO_2$ diagram (Figure 4a), the volcanic rocks is plotted in the field of basalt-andesite. The Jiangling volcanic samples have been subjected to varying degrees of alteration and have medium to relatively high loss on ignition (LOI) values of 0.51–4.88 wt.%. The samples could have affected the fluid-mobile elements, e.g., Na, K, Rb, Ba, and Sr. Therefore, we use both immobile incompatible elements (e.g., HFSE and REE) and transition metals (e.g., Sc and Co) to classify the Jiangling samples and discuss their petrogenesis. Based on the Nb/Y vs. $Zr/TiO_2$ diagram of Winchester and Floyd (1977) [62], all the analyzed samples are plotted in the alkali basalt field (Figure 4b).

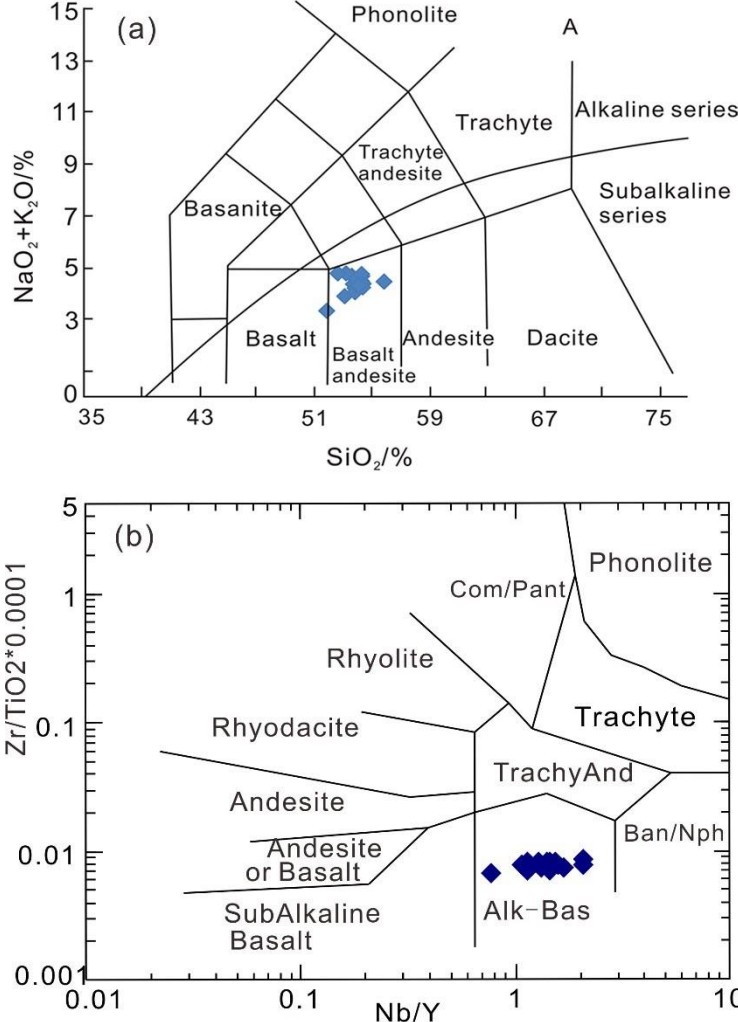

**Figure 4.** (**a**) $Na_2O + K_2O$ vs. $SiO_2$ classification diagram for volcanic rocks from the Jiangling Basin. (**b**) $Zr/TiO_2$*0.0001-Nb/Y classification diagram for volcanic rocks from the Jiangling Basin (after [63]).

**Table 1.** Major and trace element compositions of the whole volcanic rock samples from Jiangling Basin.

| Sample | B01 | B03 | B04 | B05 | B06 | B07 | BLS1 | BLS3 | BLS4 | BLS5-1 | BLS5-2 | BLS5-3 | 2062 | TW01-4 | TW02-1 | TW04-1 | TW04-2 | TW05-1 | TW06-3 | TW07-1 | TW07-2 |
|---|---|---|---|---|---|---|---|---|---|---|---|---|---|---|---|---|---|---|---|---|---|
| **Major oxides (wt%)** | | | | | | | | | | | | | | | | | | | | | |
| $SiO_2$ (%) | 53.01 | 52.45 | 53.46 | 53.1 | 53.43 | 52.8 | 53.7 | 53.24 | 53.2 | 52.96 | 50.19 | 54.27 | 50.32 | 52.9 | 52.81 | 53.09 | 51.35 | 52.42 | 51.87 | 52.56 | 52.66 |
| $TiO_2$ (%) | 1.99 | 1.98 | 1.91 | 1.89 | 1.91 | 1.9 | 1.87 | 1.95 | 1.99 | 1.92 | 1.9 | 1.95 | 1.76 | 1.97 | 2 | 1.95 | 1.99 | 1.9 | 1.88 | 1.96 | 1.96 |
| $Al_2O_3$ (%) | 14.86 | 14.89 | 14.85 | 15.17 | 15.2 | 15.27 | 14.88 | 14.9 | 14.78 | 14.6 | 15.7 | 15.17 | 13.57 | 14.94 | 15.01 | 15.04 | 15.56 | 15.06 | 14.94 | 15.24 | 15.02 |
| $Fe_2O_3$ (%) | 2.72 | 2.56 | 2.94 | 2.76 | 3.53 | 3.33 | 2.33 | 1.98 | 4.06 | 3.49 | 6.91 | 4.53 | 4.71 | 3 | 3.32 | 4.8 | 8.7 | 5.97 | 4.63 | 4.62 | 5.69 |
| FeO (%) | 7.04 | 7.17 | 6.52 | 6.48 | 5.73 | 5.87 | 6.79 | 7.26 | 5.52 | 6.11 | 2.87 | 3.52 | 5.16 | 6.43 | 5.65 | 4.67 | 1.42 | 3.07 | 4.08 | 4.4 | 3.86 |
| MnO (%) | 0.15 | 0.14 | 0.13 | 0.12 | 0.11 | 0.11 | 0.13 | 0.12 | 0.12 | 0.11 | 0.08 | 0.13 | 0.13 | 0.13 | 0.12 | 0.13 | 0.08 | 0.13 | 0.1 | 0.11 | 0.12 |
| MgO (%) | 6.36 | 6.58 | 6.3 | 6.03 | 5.81 | 5.85 | 5.7 | 5.8 | 6.74 | 6.62 | 4.07 | 6.84 | 6.05 | 5.5 | 5.44 | 5.27 | 5.13 | 5.51 | 5.43 | 5.55 | 5.55 |
| CaO (%) | 7.58 | 7.53 | 7.86 | 8.12 | 7.84 | 8.4 | 8.11 | 7.84 | 7.89 | 7.88 | 8.69 | 8.85 | 8.1 | 7.73 | 7.76 | 7.94 | 8.01 | 8.25 | 9 | 8.18 | 7.87 |
| $Na_2O$ (%) | 3.42 | 3.49 | 3.2 | 3.24 | 3.38 | 3.27 | 3.36 | 3.37 | 3.2 | 3.1 | 2.97 | 3.36 | 3.09 | 3.4 | 3.44 | 3.19 | 3.17 | 3.13 | 3.05 | 3.22 | 3.13 |
| $K_2O$ (%) | 1.19 | 1.2 | 1.04 | 1.09 | 1.01 | 1 | 0.95 | 1.19 | 1.09 | 1.02 | 0.25 | 0.96 | 1.45 | 1.06 | 1.16 | 1.06 | 0.58 | 0.94 | 0.84 | 0.91 | 1.05 |
| $P_2O_5$ (%) | 0.4 | 0.4 | 0.34 | 0.35 | 0.36 | 0.36 | 0.34 | 0.38 | 0.34 | 0.33 | 0.34 | 0.37 | 0.25 | 0.36 | 0.39 | 0.35 | 0.37 | 0.34 | 0.33 | 0.35 | 0.35 |
| $H_2O^+$ (%) | 1.1 | 0.86 | 0.66 | 0.88 | 1.5 | 1.04 | 1.34 | 1.52 | 1.42 | 1.38 | 2.9 | 1.7 | 3.32 | 1.5 | 2.23 | 2.26 | 3.72 | 3.42 | 3.32 | 2.62 | 2.92 |
| $CO_2$ (%) | 0.12 | 0.07 | 0.11 | 0.11 | 0.11 | 0.16 | 0.09 | 0.09 | 0.1 | 0.05 | 0.11 | 0.79 | 1.81 | 0.26 | 0.26 | 0.17 | 0.26 | 0.09 | 0.43 | 0.34 | 0.34 |
| LOI (%) | 0.51 | 0.53 | 0.62 | 0.75 | 0.88 | 0.49 | 0.82 | 0.73 | 0.89 | 0.51 | 2.57 | 1.89 | 4.88 | 0.97 | 1.63 | 1.73 | 3.95 | 3.26 | 3.73 | 2.41 | 2.59 |
| SUM | 98.72 | 98.39 | 98.55 | 98.35 | 98.31 | 98.16 | 98.63 | 97.93 | 97.99 | 98.26 | 96.52 | 97.18 | 95.38 | 97.97 | 97.16 | 97.66 | 96.5 | 96.34 | 96.23 | 96.98 | 97.26 |
| **Trace elements (ppm)** | | | | | | | | | | | | | | | | | | | | | |
| La | 29.3 | 28.6 | 22.4 | 21.6 | 23.5 | 21.7 | 23.4 | 27.6 | 22.5 | 20.8 | 21.7 | 22.7 | 16.1 | 25 | 27.9 | 22.3 | 22.3 | 22.3 | 20.1 | 22.1 | 22.6 |
| Ce | 56.2 | 54.5 | 43.2 | 39.5 | 44.3 | 40.8 | 42.6 | 49.1 | 42.3 | 39.5 | 41 | 42.4 | 31.5 | 45 | 50.2 | 41.3 | 40.9 | 41 | 37.7 | 41 | 41.5 |
| Pr | 5.99 | 5.7 | 4.94 | 4.73 | 4.87 | 4.74 | 5.27 | 5.87 | 5.19 | 4.95 | 5.14 | 5.23 | 4.29 | 5.68 | 6.38 | 5.51 | 5.47 | 5.51 | 4.97 | 5.48 | 5.53 |
| Nd | 25.1 | 23.1 | 21.2 | 20.9 | 20.4 | 20.9 | 21.6 | 24.4 | 22.4 | 20.8 | 22 | 22.5 | 17.8 | 22.6 | 24.3 | 22.2 | 21.7 | 22.4 | 20.5 | 22.1 | 22.4 |
| Sm | 5.28 | 5.24 | 5.18 | 4.81 | 4.58 | 4.91 | 5 | 5.38 | 5.21 | 4.79 | 4.94 | 5.08 | 4.64 | 5.58 | 5.98 | 5.7 | 5.39 | 5.56 | 5.03 | 5.53 | 5.79 |
| Eu | 1.91 | 1.75 | 1.72 | 1.71 | 1.7 | 1.77 | 1.89 | 1.94 | 1.9 | 1.76 | 1.91 | 1.91 | 1.7 | 1.89 | 2.03 | 1.93 | 1.87 | 1.93 | 1.81 | 1.88 | 1.93 |
| Gd | 6.23 | 5.65 | 5.48 | 5.57 | 5.71 | 5.66 | 5.39 | 5.77 | 5.48 | 5.11 | 5.47 | 5.61 | 4.97 | 5.41 | 5.79 | 5.52 | 5.35 | 5.5 | 5.04 | 5.39 | 5.58 |
| Tb | 0.98 | 0.87 | 0.85 | 0.83 | 0.82 | 0.86 | 0.79 | 0.88 | 0.79 | 0.78 | 0.79 | 0.82 | 0.73 | 0.74 | 0.81 | 0.76 | 0.76 | 0.78 | 0.72 | 0.79 | 0.77 |
| Dy | 4.6 | 4.23 | 4.09 | 3.93 | 3.86 | 3.89 | 4.24 | 4.61 | 4.32 | 4.06 | 4.19 | 4.4 | 4.22 | 4.29 | 4.53 | 4.31 | 4.27 | 4.3 | 3.98 | 4.31 | 4.25 |
| Ho | 0.84 | 0.79 | 0.75 | 0.74 | 0.71 | 0.76 | 0.75 | 0.83 | 0.81 | 0.75 | 0.77 | 0.81 | 0.81 | 0.77 | 0.83 | 0.79 | 0.74 | 0.76 | 0.71 | 0.78 | 0.78 |
| Er | 2.39 | 2.28 | 2.17 | 2.13 | 2.01 | 2.12 | 1.9 | 2.12 | 1.97 | 1.82 | 1.94 | 2 | 2.28 | 2.1 | 2.24 | 2.12 | 2.07 | 2.12 | 2 | 2.12 | 2.17 |
| Tm | 0.29 | 0.24 | 0.23 | 0.23 | 0.25 | 0.22 | 0.27 | 0.29 | 0.28 | 0.25 | 0.26 | 0.26 | 0.27 | 0.25 | 0.28 | 0.26 | 0.25 | 0.26 | 0.24 | 0.25 | 0.26 |
| Yb | 1.73 | 1.69 | 1.6 | 1.68 | 1.54 | 1.57 | 1.36 | 1.59 | 1.5 | 1.43 | 1.51 | 1.62 | 1.77 | 1.56 | 1.63 | 1.58 | 1.57 | 1.55 | 1.45 | 1.55 | 1.66 |
| Lu | 0.24 | 0.24 | 0.22 | 0.2 | 0.22 | 0.22 | 0.2 | 0.24 | 0.22 | 0.2 | 0.21 | 0.21 | 0.25 | 0.22 | 0.24 | 0.23 | 0.21 | 0.21 | 0.21 | 0.22 | 0.23 |
| Y | 21.8 | 20.5 | 19.2 | 20.2 | 19.7 | 19.6 | 17.9 | 20.5 | 20.5 | 18.1 | 18.9 | 20.1 | 21.8 | 20.7 | 23.2 | 21.9 | 20.9 | 21.1 | 19.3 | 21 | 21 |
| Ba | 357 | 349 | 282 | 277 | 284 | 304 | 288 | 316 | 292 | 280 | 205 | 297 | 232 | 289 | 325 | 281 | 288 | 293 | 255 | 271 | 301 |
| Rb | 32.5 | 31.8 | 25.4 | 26.5 | 26 | 20.1 | 23.5 | 32.2 | 25.6 | 22.3 | 2.43 | 15.7 | 10.5 | 26.5 | 28.8 | 22.5 | 3.9 | 17.3 | 13.2 | 14.8 | 23.1 |
| Sr | 595 | 548 | 456 | 467 | 453 | 561 | 433 | 450 | 446 | 434 | 500 | 467 | 408 | 508 | 658 | 471 | 486 | 469 | 494 | 464 | 452 |
| Ta | 1.92 | 1.77 | 1.43 | 1.36 | 1.45 | 1.35 | 1.46 | 1.69 | 1.49 | 1.41 | 1.45 | 1.47 | 1.15 | 1.56 | 1.72 | 1.4 | 1.42 | 1.43 | 1.28 | 1.43 | 1.43 |
| Nb | 40.7 | 38.3 | 29 | 29.2 | 31.9 | 28.6 | 26.1 | 32 | 27.8 | 24.6 | 25.9 | 27.3 | 20.5 | 29.9 | 34.1 | 26 | 26.2 | 25.9 | 24 | 25.5 | 26.3 |
| Hf | 4.07 | 3.84 | 3.73 | 3.65 | 3.56 | 3.7 | 3.84 | 4.37 | 4.28 | 3.99 | 4.15 | 4.35 | 3.51 | 4.06 | 4.26 | 4.19 | 4 | 4.27 | 3.84 | 4.08 | 4.22 |
| Zr | 170 | 162 | 157 | 160 | 150 | 157 | 138 | 160 | 161 | 147 | 154 | 162 | 132 | 152 | 168 | 159 | 155 | 160 | 146 | 155 | 158 |
| Th | 4.01 | 3.92 | 2.79 | 2.82 | 3.14 | 2.86 | 3.36 | 4.02 | 3.37 | 3.18 | 3.35 | 3.56 | 2.53 | 3.7 | 3.95 | 3.11 | 3.19 | 3.22 | 3.01 | 3.29 | 3.26 |
| U | 0.96 | 0.87 | 0.7 | 0.68 | 0.71 | 0.68 | 0.72 | 0.78 | 0.6 | 0.64 | 0.68 | 0.81 | 0.65 | 0.7 | 0.79 | 0.56 | 0.62 | 0.63 | 0.61 | 0.57 | 0.65 |
| Cs | 0.33 | 0.32 | 0.22 | 0.27 | 0.95 | 0.1 | 0.21 | 1.34 | 0.29 | 0.21 | 0.13 | 0.19 | 0.6 | 0.25 | 0.3 | 0.22 | 0.1 | 0.2 | 0.1 | 0.17 | 0.25 |
| Sc | 22 | 20.7 | 21.1 | 20.7 | 20.7 | 21.6 | 16 | 18.6 | 20 | 18.8 | 19 | 19.8 | 23.4 | 21.2 | 22.5 | 21.7 | 20.5 | 21.9 | 20.3 | 20.9 | 21.6 |
| Co | 42.4 | 40.5 | 39.7 | 40 | 40.5 | 38.9 | 35.6 | 38 | 38 | 37.9 | 36.5 | 30.9 | 38.1 | 36.7 | 38.1 | 36.6 | 32.9 | 33.9 | 35.4 | 36.9 | 38.8 |
| Cu | 61.3 | 55 | 37 | 37.8 | 54.1 | 45.6 | 47.9 | 54.5 | 37 | 40.6 | 49.7 | 52.1 | 60 | 52.6 | 57.1 | 33.6 | 44.5 | 35.7 | 74.7 | 48.9 | 41.6 |
| Cr | 218 | 197 | 195 | 199 | 197 | 200 | 159 | 168 | 178 | 166 | 172 | 162 | 196 | 181 | 197 | 191 | 183 | 190 | 190 | 187 | 190 |
| V | 159 | 150 | 143 | 141 | 141 | 144 | 119 | 137 | 138 | 126 | 129 | 134 | 156 | 135 | 148 | 137 | 119 | 144 | 133 | 130 | 139 |
| Ni | 144 | 143 | 128 | 134 | 133 | 127 | 122 | 132 | 132 | 132 | 127 | 96.4 | 131 | 128 | 119 | 141 | 104 | 132 | 122 | 130 | 131 |
| $Eu/Eu^*$ | 1.016 | 0.9777 | 0.98035 | 1.007309 | 1.01542 | 1.023584 | 1.1066974 | 1.0581653 | 1.07947 | 1.08079259 | 1.1183601 | 1.088862976 | 1.07575 | 1.03854836 | 1.04145711 | 1.0387052 | 1.05320118 | 1.05529313 | 1.08795066 | 1.04005937 | 1.02465546 |
| La/Yb | 11.4448 | 11.436 | 9.46049 | 8.688206 | 10.3118 | 9.339975 | 11.626863 | 11.729988 | 10.1362 | 9.829081 | 9.71109948 | 9.468833047 | 6.14665 | 10.829316 | 11.5665067 | 9.5374746 | 9.59822281 | 9.72207084 | 9.36728366 | 9.63487738 | 9.19996061 |

Abbreviations: $Eu/Eu^* = Eu_N/[(Sm_N + Gd_N)/2]$.

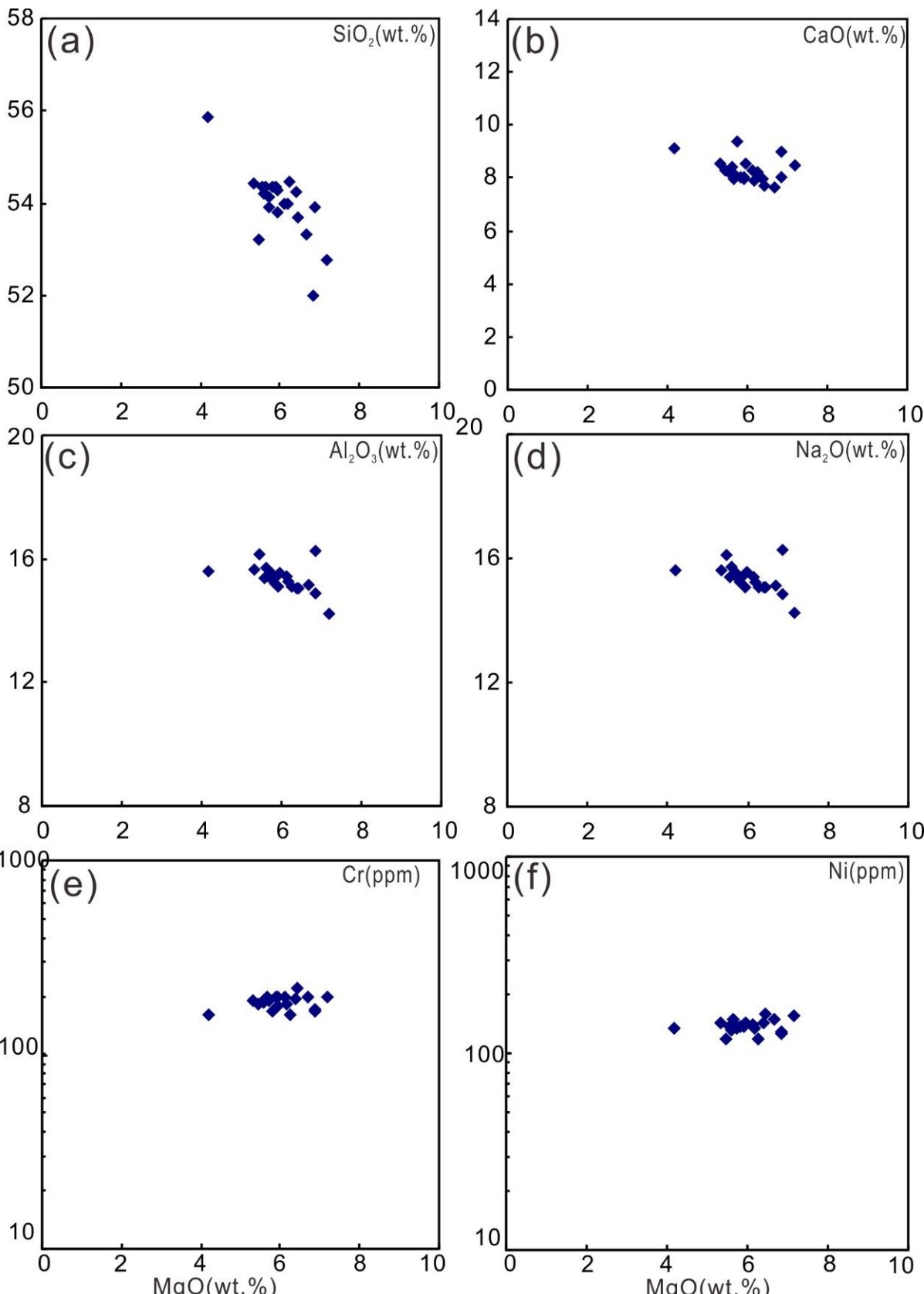

**Figure 5.** Harker diagrams showing the whole-rock compositional range of volcanic rocks from the Jiangling Basin. (**a**) contents of $SiO_2$. (**b**) contents of CaO. (**c**) contents of $Al_2O_3$. (**d**) contents of $Na_2O$. (**e**) contents of Cr. (**f**) contents of Ni.

The volcanic rocks show similar chondrite-normalized REE patterns (Figure 6a), with OIB-like enrichments of 10 times (HREE) to 100 times (LREE) over chondrite [64]. The rocks have $(La/Yb)_N$ ratios of 6.14–11.7 (N: chondrite normalized), and they show no Eu anomalies, with Eu/Eu* values of 0.98–1.09. On the primitive mantle-normalized multi-element diagram (Figure 6b), the samples are highly enriched in large-ion lithophile elements (LILE) and Th, and the HFSE, Nb and Ta exhibit negative to positive anomalies similar to oceanic island basalts (OIB) [65]. The composition of the samples is also similar to volcanic from the southern margin of the North China block, as observed in the initial geochemical analysis of three basaltic rocks [66]. Relevant incompatible element ratios are Nb/La = 1.15–1.39, Th/La = 0.12–0.16, Th/Nb = 1.43–2.53, and Nb/U = 31.5–46.4.

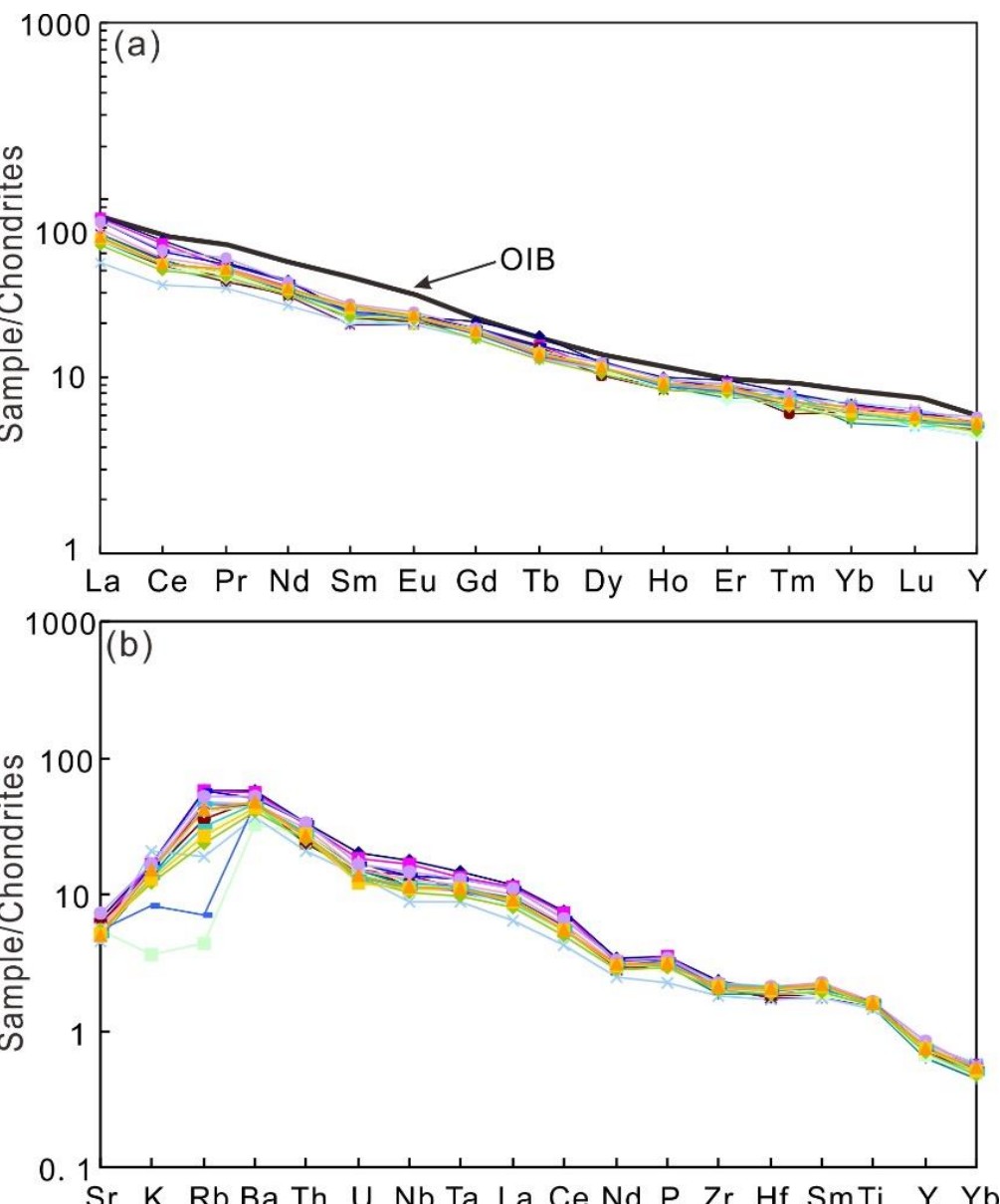

**Figure 6.** (**a**) Chondrite-normalized REEs of volcanic rocks from the Jiangling Basin (after [64]). (**b**) Spider diagram patterns of volcanic rocks from the Jiangling Basin (after [64]). OIB-Oceanic Island Basalt.

### 4.2. $^{40}Ar/^{39}Ar$ Dating

For each sample, we calculated: (1) a plateau age, which was evaluated from the apparent age spectrum of the progressive heating steps according to the statistical criteria

set out in McDougall and Harrison (1988) [67]; and (2) a normal isochron age, which was calculated from a regression analysis of $^{36}Ar/^{40}Ar$ vs. $^{39}Ar/^{40}Ar$ data using the Isoplot 3.0 program of Ludwig (2003) [68]. A summary of the $^{40}Ar/^{39}Ar$ age results for the Jiangling Basin obtained during this study is presented in Table 2 and Figures 7 and 8.

**Table 2.** $^{40}Ar/^{39}Ar$ isotopic analytical data of the incremental heating experiments on basaltic rocks.

| Sample | Temperature (°C) | $(^{40}Ar/^{39}Ar)_m$ | $(^{36}Ar/^{39}Ar)_m$ | $(^{37}Ar/^{39}Ar)_m$ | $^{40}Ar^*$ (%) | F $(^{40}Ar^*/^{39}Ar)$ | $^{39}Ar$ (%) | Age (Ma) | $\pm 1\sigma$ (Ma) |
|---|---|---|---|---|---|---|---|---|---|
| | 600 | 851.6401 | 2.6048 | 0.4219 | 9.62 | 81.8618 | 0.83 | 606 | 11 |
| | 670 | 74.1470 | 0.1192 | 1.1909 | 52.62 | 39.0576 | 2.25 | 314.5 | 5.6 |
| | 810 | 30.3102 | 0.0729 | 1.1422 | 29.19 | 8.8566 | 19.37 | 76.3 | 3.7 |
| B01 | 940 | 7.4940 | 0.0044 | 1.3823 | 84.15 | 6.3137 | 51.96 | 54.70 | 0.35 |
| | 1060 | 16.8665 | 0.0322 | 0.8854 | 43.98 | 7.4243 | 20.19 | 64.1 | 1.7 |
| | 1300 | 8.1117 | 0.0163 | 34.8969 | 75.91 | 6.3462 | 3.99 | 55.0 | 1.1 |
| | 1420 | 9.0511 | 0.0124 | 22.3723 | 79.71 | 7.3551 | 1.41 | 63.6 | 1.1 |
| | 600 | 278.2450 | 0.8290 | 1.1535 | 11.99 | 33.4037 | 1.38 | 272 | 38 |
| | 670 | 14.5806 | 0.0323 | 0.2648 | 34.69 | 5.0587 | 16.40 | 43.9 | 1.7 |
| | 740 | 15.9424 | 0.0307 | 1.5654 | 43.81 | 6.9945 | 25.58 | 60.4 | 1.6 |
| B02 | 810 | 11.1885 | 0.0155 | 0.9881 | 59.74 | 6.6902 | 23.93 | 57.84 | 0.84 |
| | 940 | 14.7641 | 0.0238 | 0.4479 | 52.60 | 7.7692 | 16.74 | 67.0 | 1.3 |
| | 1060 | 11.6372 | 0.0076 | 0.9400 | 81.44 | 9.4848 | 13.10 | 81.46 | 0.56 |
| | 1320 | 9.4148 | 0.0125 | 20.8622 | 79.17 | 7.5890 | 2.87 | 65.47 | 0.84 |
| | 600 | 274.0982 | 0.8290 | 1.1535 | 10.66 | 29.2528 | 1.37 | 240 | 39 |
| | 670 | 14.9305 | 0.0323 | 0.2648 | 36.22 | 5.4086 | 16.24 | 46.9 | 1.7 |
| | 740 | 15.9576 | 0.0308 | 1.6279 | 43.72 | 6.9869 | 27.84 | 60.4 | 1.6 |
| B03 | 810 | 11.1223 | 0.0150 | 0.6251 | 60.53 | 6.7360 | 23.81 | 58.23 | 0.82 |
| | 940 | 14.7508 | 0.0237 | 0.3904 | 52.70 | 7.7757 | 14.93 | 67.0 | 1.3 |
| | 1060 | 11.6372 | 0.0076 | 0.9400 | 81.44 | 9.4848 | 12.97 | 81.46 | 0.56 |
| | 1320 | 9.4148 | 0.0125 | 20.8622 | 79.17 | 7.5890 | 2.85 | 65.47 | 0.84 |
| | 740 | 30.6672 | 0.0823 | 1.2765 | 21.02 | 6.45209 | 1.47 | 55.8 | 4.3 |
| | 810 | 14.2521 | 0.0245 | 0.9374 | 49.77 | 7.09852 | 21.49 | 61.3 | 1.3 |
| | 940 | 7.7797 | 0.0056 | 1.7824 | 80.61 | 6.28089 | 47.43 | 54.37 | 0.40 |
| B04 | 1060 | 8.1126 | 0.0069 | 1.4887 | 76.50 | 6.21359 | 18.66 | 53.80 | 0.45 |
| | 1180 | 9.6164 | 0.0129 | 4.1482 | 63.89 | 6.16535 | 7.38 | 53.39 | 0.72 |
| | 1320 | 16.3593 | 0.0411 | 32.5171 | 42.11 | 7.08441 | 1.56 | 61.2 | 2.3 |
| | 1400 | 19.1049 | 0.0490 | 31.7098 | 37.83 | 7.42885 | 1.99 | 64.1 | 2.7 |
| | 600 | 684.8268 | 2.0468 | 1.3359 | 11.70 | 80.1872 | 0.93 | 595 | 155 |
| | 670 | 10.5382 | 0.0132 | 0.9222 | 63.79 | 6.7272 | 12.60 | 58.2 | 3.1 |
| | 810 | 8.5275 | 0.0081 | 1.6674 | 73.58 | 6.2832 | 52.10 | 54.4 | 1.0 |
| B05 | 940 | 7.0973 | 0.0035 | 1.8539 | 87.46 | 6.2170 | 20.55 | 53.82 | 0.88 |
| | 1060 | 9.1753 | 0.0142 | 1.8734 | 55.81 | 5.1285 | 8.26 | 44.5 | 1.5 |
| | 1180 | 7.2295 | 0.0235 | 22.1128 | 29.33 | 2.1609 | 2.20 | 18.9 | 2.7 |
| | 1320 | 9.0652 | 0.0263 | 26.4668 | 38.16 | 3.5392 | 3.35 | 30.8 | 2.9 |
| | 600 | 371.4065 | 1.1348 | 5.2664 | 9.83 | 36.6825 | 2.15 | 296.49 | 103 |
| | 670 | 31.4003 | 0.0796 | 0.7308 | 25.29 | 7.9469 | 8.81 | 68.5 | 8.1 |
| | 810 | 24.6692 | 0.0527 | 1.2800 | 37.31 | 9.2143 | 39.43 | 79.2 | 5.4 |
| B06 | 940 | 10.4640 | 0.0149 | 2.9990 | 60.20 | 6.3150 | 27.39 | 54.6 | 1.6 |
| | 1060 | 20.8621 | 0.0470 | 2.0688 | 34.32 | 7.1732 | 11.29 | 61.9 | 4.9 |
| | 1180 | 10.7466 | 0.0208 | 15.5130 | 54.57 | 5.9428 | 9.37 | 51.5 | 2.3 |
| | 1320 | 10.8196 | 0.0262 | 34.3771 | 54.69 | 6.0958 | 1.57 | 52.8 | 3.5 |
| | 600 | 360.8878 | 1.0698 | 1.0411 | 12.42 | 44.87568 | 1.58 | 357.03 | 46.36 |
| | 670 | 9.0914 | 0.0125 | 1.0297 | 60.29 | 5.48576 | 15 | 47.62 | 0.69 |
| B07 | 810 | 7.6023 | 0.0053 | 1.7737 | 81.4 | 6.19804 | 56.54 | 53.71 | 0.38 |
| | 940 | 7.0516 | 0.0038 | 2.1855 | 86.51 | 6.11198 | 20.55 | 52.98 | 0.33 |
| | 1060 | 9.8447 | 0.0135 | 3.005 | 62.11 | 6.1303 | 4.91 | 53.13 | 0.77 |
| | 1180 | 9.0463 | 0.0212 | 34.0492 | 61.72 | 5.75003 | 1.41 | 49.88 | 1.51 |
| | 600 | 387.1796 | 1.1614 | 0.0999 | 11.37 | 44.0084 | 1.16 | 350 | 51 |
| | 630 | 16.3555 | 0.0466 | 0.2382 | 15.90 | 2.6003 | 6.27 | 22.7 | 2.4 |
| | 670 | 12.9349 | 0.0280 | 0.2795 | 36.31 | 4.6977 | 21.73 | 40.8 | 1.5 |
| | 740 | 12.5528 | 0.0220 | 0.2645 | 48.38 | 6.0746 | 25.24 | 52.5 | 1.2 |
| B08 | 810 | 14.9591 | 0.0304 | 0.1403 | 40.01 | 5.9853 | 14.16 | 51.8 | 1.6 |
| | 860 | 13.5351 | 0.0243 | 0.1546 | 46.99 | 6.3615 | 7.75 | 55.0 | 1.3 |
| | 1000 | 8.9940 | 0.0093 | 0.0721 | 69.37 | 6.2399 | 8.84 | 53.96 | 0.56 |
| | 1060 | 13.3248 | 0.0227 | 0.1182 | 49.75 | 6.6302 | 7.55 | 57.3 | 1.2 |
| | 1180 | 11.2519 | 0.0077 | 0.2064 | 79.84 | 8.9854 | 7.32 | 77.20 | 0.57 |

Note: Ar* represents radiogenic $^{40}Ar$ from the decay of $^{40}K$.

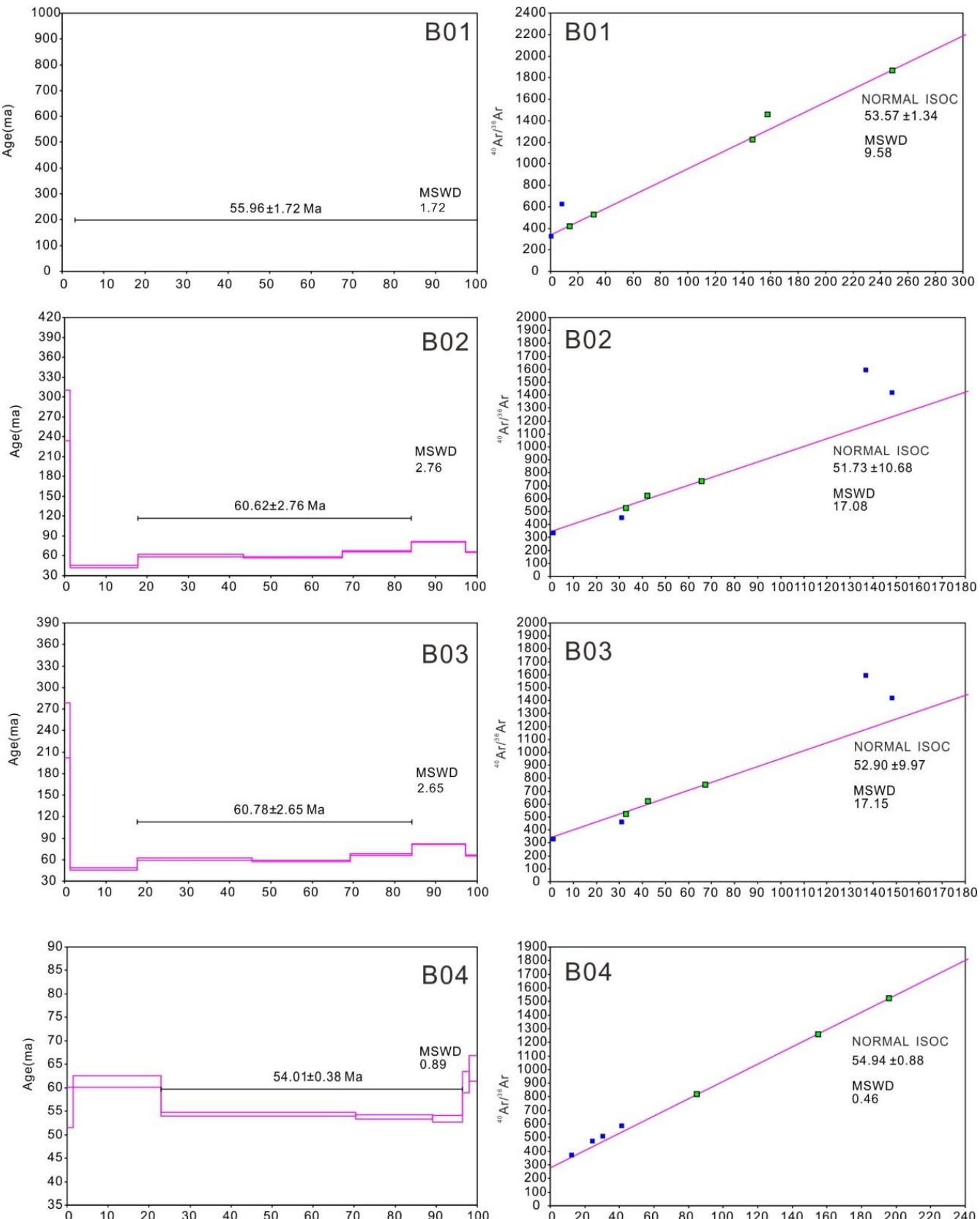

**Figure 7.** $^{40}$Ar/$^{39}$Ar age spectra and isochron ages (Sample B01–B04) of the volcanic rocks from the Jiangling Basin.

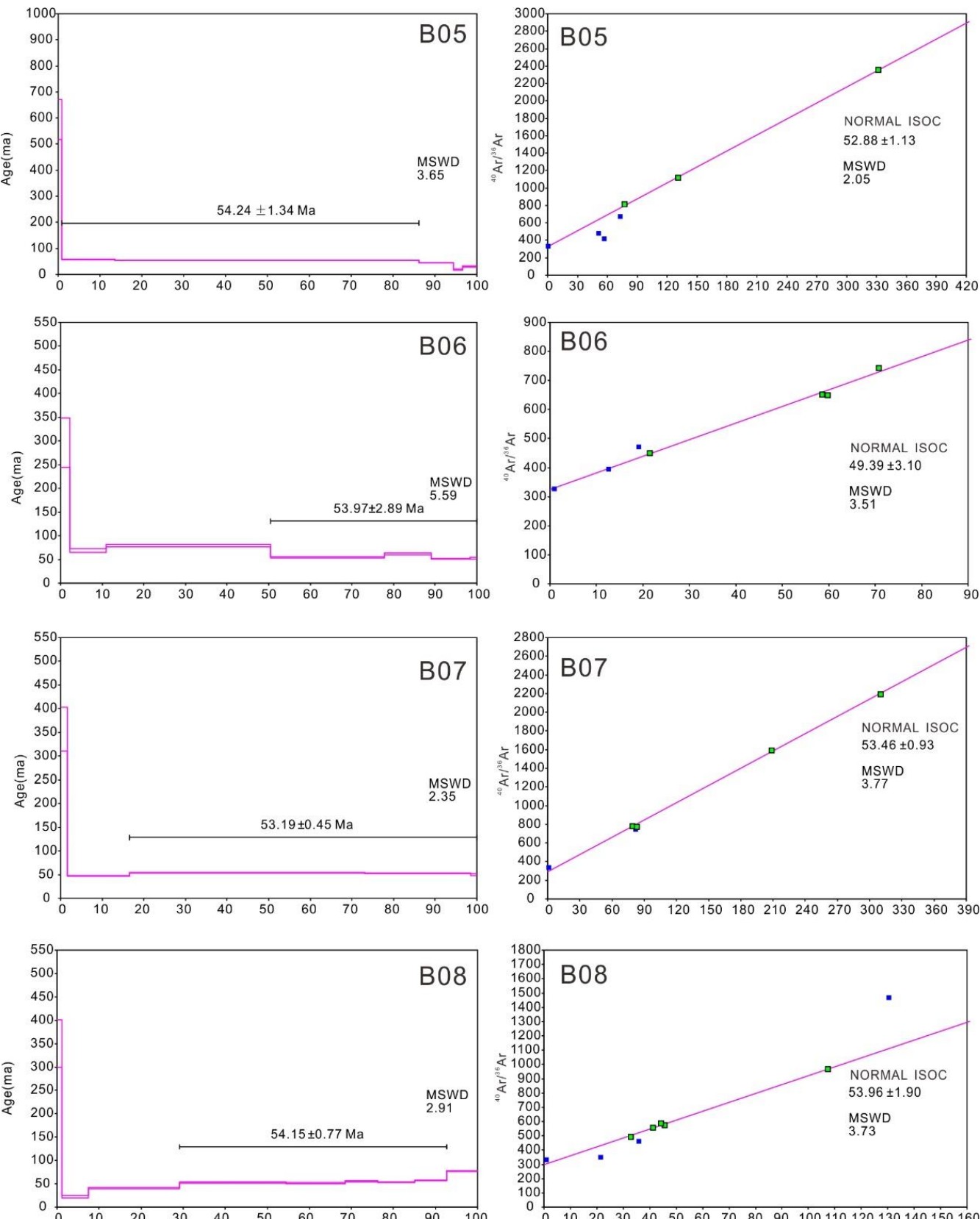

**Figure 8.** $^{40}Ar/^{39}Ar$ age spectra and isochron ages (Sample B05–B08) of the volcanic rocks from the Jiangling Basin.

The groundmass argon release patterns of most experiments showed consistent results with plateaus that met commonly accepted reliability criteria (McDougall and Harrison, 1988). The mean squares weighted deviates (MSWD) values were used to define the plateau segments [69–71]. All samples yielded plateau segments with MSWD < 5.59, indicating that the gas was derived from one isotopically homogeneous reservoir. In addition, the obtained plateau ages consisted of more than three steps and more than 50% of the total gas was released, which O'Connor et al. (2007) [72] noted provided the best representation of the eruption age of individual flows. Several spectra showed elevated ages in the initial and final incremental heating steps. Elevated ages in the initial steps may indicate either loosely bound excess $^{40}$Ar or to the recoil loss of $^{39}$Ar from fine-grained alteration phases [73]. The elevated ages in the final temperature steps may be related to the admixture of phenocryst-hosted inherited $^{40}$Ar [73–77]. Such elevated results at the beginning or end of the experiments were not included in the plateau age calculations.

When the data from individual experiments were regressed in isochrons, the dispersion of points was often too small to calculate reliable regression lines. The $^{40}$Ar/$^{36}$Ar intercept at the 95% confidence level was indistinguishable from the $^{40}$Ar/$^{36}$Ar of air within the uncertainty limits. This finding indicates that the groundmass of the erupted rocks was essentially free of extraneous $^{40}$Ar. However, in many samples, the uncertainties in the isochron age calculations were high, and the isochrons yielded lower-precision ages than plateau ages.

The samples exhibited typical plateau ages without the heat disturbance of an open system. For example, in sample B01, the low-temperature heating step (600–810°C) plateau age of 606 ± 11 Ma rapidly decreased to 54.7 ± 0.35 Ma when heating to 1420°C; the cumulative release of $^{39}$Ar during high-temperature heating was 17%. The seven continuous $^{40}$Ar/$^{39}$Ar plateau ages of the heating steps produced a consistent spectrum of ages, and the cumulative amount of $^{39}$Ar was 79.71%. The plateau age of the sample was 55.96 ± 1.72 Ma, which is similar to the isochron age within error (53.57 ± 1.34 Ma) (with a higher MSWD of 9.58). The initial value of 294.4 $^{40}$Ar/$^{39}$Ar was similar to the pressure value of 295.5, indicating that these samples were not influenced by excess argon micro-cracks in gas operations [78]. Therefore, the plateau age is reliable and represents the magma cooling age. All the other samples have the same reliable data. Because the samples were not affected by deformation and metamorphism, we interpret the obtained plateau ages as emplacement ages, and the age of the Hanjiang River Basin and the Shashi group is as early as the Tertiary volcanic rocks with similar K-Ar ages [36,56].

### 4.3. Sr-Nd-Pb Isotopes

The $^{87}$Sr/$^{86}$Sr isotopic ratios and $\varepsilon_{Nd(t)}$ values for the Jiangling volcanic rocks range from 0.7043 to 0.7072 and +2.38 to −2.22, respectively (Table 3). The Pb isotopic ratios range from 16.9 to 18.0 of $^{206}$Pb/$^{204}$Pb, 15.3 to 15.5 of $^{207}$Pb/$^{204}$Pb, and 37.1 to 38.1 of $^{208}$Pb/$^{204}$Pb (Figure 9). The Jiangling volcanic rocks lie along mixing trends defined by the EMI and mid-ocean ridge basalt (MORB) components in the $^{143}$Nd/$^{144}$Nd and $^{87}$Sr/$^{86}$Sr vs. $^{206}$Pb/$^{204}$Pb diagrams and the $^{87}$Sr/$^{86}$Sr ratios are low. In contrast, the EMI-like composition has been characterized by low $^{143}$Nd/$^{144}$Nd and $^{206}$Pb/$^{204}$Pb ratios and moderately high $^{87}$Sr/$^{86}$Sr ratios.

**Table 3.** Sr-Nd-Pb isotopic compositions of the Jiangling basaltic rocks.

| Sample | $^{87}$Sr/$^{86}$Sr | ±1σ | $^{143}$Nd/$^{144}$Nd | ±1σ | $^{206}$Pb/$^{204}$Pb | ±1σ | $^{207}$Pb/$^{204}$Pb | ±1σ | $^{208}$Pb/$^{204}$Pb | ±1σ | $(^{206}$Pb/$^{204}$Pb$)_i$ | $(^{207}$Pb/$^{204}$Pb$)_i$ | $(^{208}$Pb/$^{204}$Pb$)_i$ | $(^{87}$Sr/$^{86}$Sr$)_i$ | εSr(0) | εSr(t) | εNd(0) | εNd(t) |
|---|---|---|---|---|---|---|---|---|---|---|---|---|---|---|---|---|---|---|
| 1-1 | 0.705217 | 9 | 0.512848 | 8 | 18.253 | 13 | 15.463 | 17 | 38.149 | 14 | 18.142 | 15.458 | 38.337 | 0.70517 | 10.2 | 10.5 | 2.17 | 1.99 |
| 1-2 | 0.705209 | 8 | 0.512814 | 5 | 18.198 | 15 | 15.497 | 15 | 38.058 | 16 | 18.103 | 15.492 | 32.101 | 0.705165 | 10.06 | 10.44 | 0.72 | 0.75 |
| 1-3 | 0.705243 | 6 | 0.512853 | 9 | 18.224 | 16 | 15.475 | 16 | 38.127 | 12 | 18.132 | 15.471 | 33.896 | 0.705197 | 10.55 | 10.89 | −0.64 | −0.82 |
| 1-4 | 0.705237 | 7 | 0.512842 | 7 | 18.201 | 13 | 15.512 | 18 | 38.075 | 13 | 18.118 | 15.508 | 33.426 | 0.705192 | 10.46 | 10.82 | 0.16 | 0.08 |
| 2-1 | 0.704295 | 9 | 0.512843 | 8 | 18.013 | 18 | 15.459 | 11 | 37.982 | 12 | 17.964 | 15.457 | 33.389 | 0.704263 | −2.91 | −2.36 | 0.86 | 0.92 |
| 2-2 | 0.704308 | 8 | 0.512856 | 6 | 18.082 | 16 | 15.403 | 15 | 37.956 | 15 | 18.031 | 15.401 | 35.03 | 0.704276 | −2.73 | −2.18 | 1.58 | 1.62 |
| 2-3 | 0.704281 | 7 | 0.512824 | 8 | 18.054 | 18 | 15.428 | 16 | 37.974 | 17 | 17.998 | 15.425 | 33.365 | 0.70425 | −3.11 | −2.54 | 1.81 | 1.95 |
| 2-4 | 0.704312 | 8 | 0.512831 | 7 | 18.029 | 13 | 15.416 | 17 | 37.983 | 17 | 17.982 | 15.414 | 37.926 | 0.704279 | −2.67 | −2.13 | 0.43 | 0.51 |
| 3-1 | 0.706487 | 9 | 0.512765 | 7 | 18.418 | 17 | 15.562 | 15 | 38.309 | 15 | 18.294 | 15.556 | 33.509 | 0.706443 | 28.2 | 28.59 | 0.62 | 0.61 |
| 3-2 | 0.706492 | 6 | 0.512756 | 8 | 18.256 | 15 | 15.507 | 16 | 38.148 | 19 | 18.163 | 15.503 | 34.734 | 0.706447 | 28.28 | 28.64 | −2.17 | −2.22 |
| 3-3 | 0.706485 | 9 | 0.512803 | 9 | 18.329 | 14 | 15.528 | 14 | 38.207 | 19 | 18.207 | 15.522 | 37.995 | 0.70644 | 28.2 | 28.5 | 2.17 | 1.97 |
| 3-4 | 0.706392 | 8 | 0.512779 | 6 | 18.291 | 17 | 15.541 | 17 | 38.263 | 15 | 18.156 | 15.535 | 36.142 | 0.706348 | 26.86 | 27.24 | 2.75 | 2.7 |
| 5-1 | 0.707129 | 7 | 0.512703 | 9 | 18.527 | 10 | 15.625 | 15 | 38.512 | 17 | 18.435 | 15.621 | 35.689 | 0.70694 | 37.32 | 35.64 | 1.27 | 1.37 |
| 5-2 | 0.707238 | 9 | 0.512732 | 8 | 18.415 | 18 | 15.558 | 16 | 38.327 | 10 | 18.309 | 15.553 | 38.188 | 0.70703 | 38.9 | 36.9 | 2.17 | 2.33 |
| 5-3 | 0.707395 | 7 | 0.512749 | 5 | 18.473 | 10 | 15.584 | 16 | 38.445 | 17 | 18.368 | 15.579 | 36.472 | 0.707185 | 41.09 | 39.12 | 2.17 | 2.25 |
| 5-4 | 0.707165 | 8 | 0.512728 | 6 | 18.436 | 18 | 15.601 | 15 | 38.486 | 16 | 18.336 | 15.596 | 37.035 | 0.70697 | 37.8 | 36.1 | 2.17 | 2.38 |

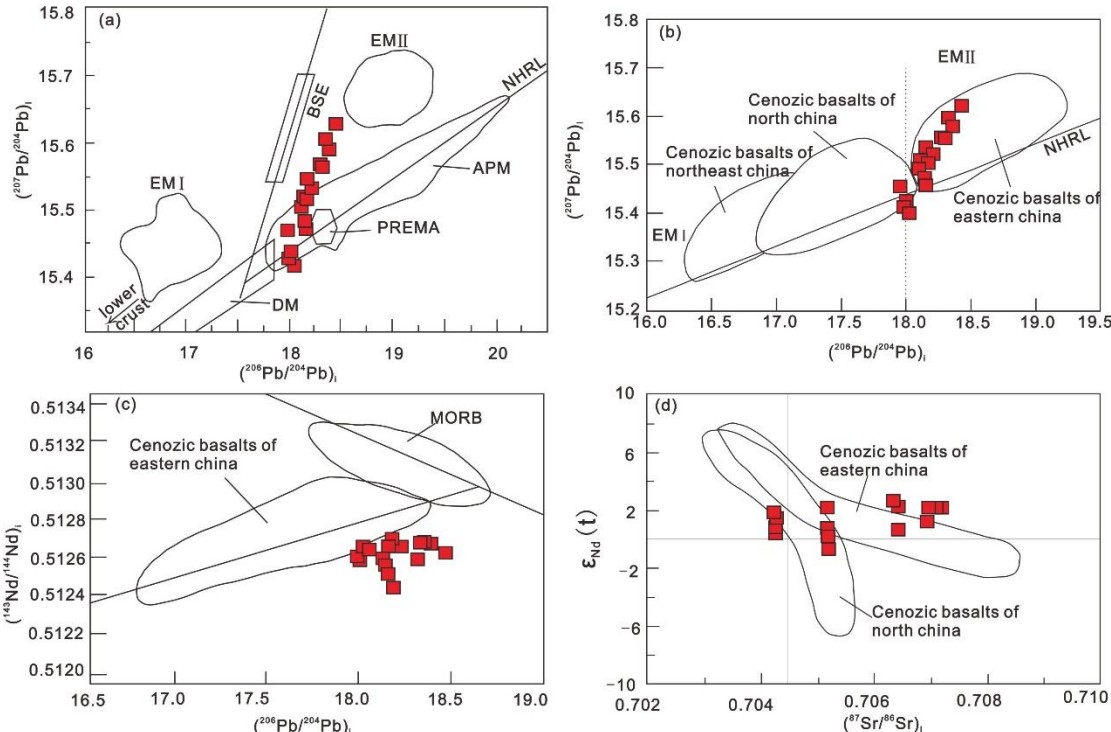

**Figure 9.** Plots of $^{207}Pb/^{204}Pb(i)$ vs. $^{206}Pb/^{204}Pb(i)$ (**a**,**b**), $^{143}Nd/^{144}Nd(i)$ vs. $^{206}Pb/^{204}Pb(i)$ (**c**) and $\varepsilon_{Nd}(t)$ vs. $^{87}Sr/^{86}Sr$ ratios (**d**) of the volcanic rocks from the Jiangling Basin. Cenozoic basalts of Eastern China from [49,50]; Cenozoic basalts of Northern China from [28,36]; Cenozoic basalts of Northeast China from [36]; EMI and EMII and DMM from [79].

## 5. Discussion

### 5.1. Source of the Volcanic Rocks

Before discussing the petrogenesis of the volcanic rocks, the low-temperature alterations and crustal or lithospheric contamination must be considered for the studied continental intraplate environment. Petrographic observations indicated that certain samples had been affected by chlorination, and the LOI values were generally low but reached 4.99% for several samples. The isotopic composition was variable, indicating that the basic rocks from the Jianghan Basin had been affected by low-temperature alteration [56]. Therefore, we focus the petrogenetic discussion on elements that are not easily affected by alteration, such as REE, HFSE, U, Th, and their ratios, such as Nb/Th, Nb/U, Th/La, Nb/La, and Zr/Nb.

The $Na_2O + K_2O$ vs. $SiO_2$ (TAS) and Nb/Y vs. $Zr/TiO_2$ diagrams of the volcanic show that all the samples are plotted in the same alkali basalt field (Figure 4). The multi-element diagram shows positive HFSEs anomalies (Figure 6), such as Nb/Ta and Nb/La, so that the volcanic are similar to the results of Peng et al. (2006) for the volcanic rocks of the Jianghan Basin, indicating that they were not contaminated by later crustal materials. The Nb/U ratio (31.5–46.3) is similar to that of MORB and OIB ($47 \pm 10$) [65] and apparently higher than the continental crust Nb/U ratio, which shows that these volcanic rocks are not affected by crustal materials and the properties of the genesis. In addition, the geochemistry and $\varepsilon_{Nd(t)}$ composition (<4.0) showed that the basic rocks from Jianghan Basin were less likely affected by lithospheric contamination.

The Paleogene volcanic rocks from Jiangling Basin have similar incompatible trace element ratios and Sr-Nd values to some of the Paleogene volcanic rocks from the southern margin of the North China Block [33,56]. The rocks from the Jiangling Basin also have variable $SiO_2$ and high $Na_2O$ values; relatively low $P_2O_5/Al_2O_3$ (0.022–0.027) and $CaO/Al_2O_3$ (0.51–0.60) ratios that exhibit relatively small changes; and Zr/Nb (4.17–6.44) and Ce/Y (1.44–2.58) ratios that are plotted in the source area of spinel peridotite. Thus, Jiangling

volcanic rocks have relatively low $(La/Yb)_N$ (9.19–11.44) and $(Gd/Yb)_N$ (2.72–3.21) ratios and relatively high HREEs content (more than ten times the chondrite-normalized values), and they belong to the alkaline group, which indicates that this basaltic magma may be the result of high-degree partial melting of spinel peridotite. On the multi-element diagram (Figure 6), all of the samples have features similar to those of OIB, such as Th/La = 0.12–0.16 and Th/Nb = 0.09–0.13, indicating that these rocks are similar to Tertiary basic rocks from the southern margin of the North China Block [66]. Therefore, these rocks may have come from an enriched lithospheric mantle with an EMII-type component.

In trace element discrimination diagrams for tectonic environments (Figure 10), all of the samples are plotted within the basaltic and enriched lithospheric mantle fields, and they have trends of intraplate enrichment [80]; therefore, the genesis of these basaltic rocks is related to the asthenospheric mantle. The upwelling of the asthenosphere resulted in the large-scale partial melting of spinel peridotite, and this process was caused by lithosphere-asthenosphere interactions. This history is similar to that of the Cenozoic basaltic rocks from the Southeastern coastal areas, which are also the result of lithosphere-asthenosphere interactions.

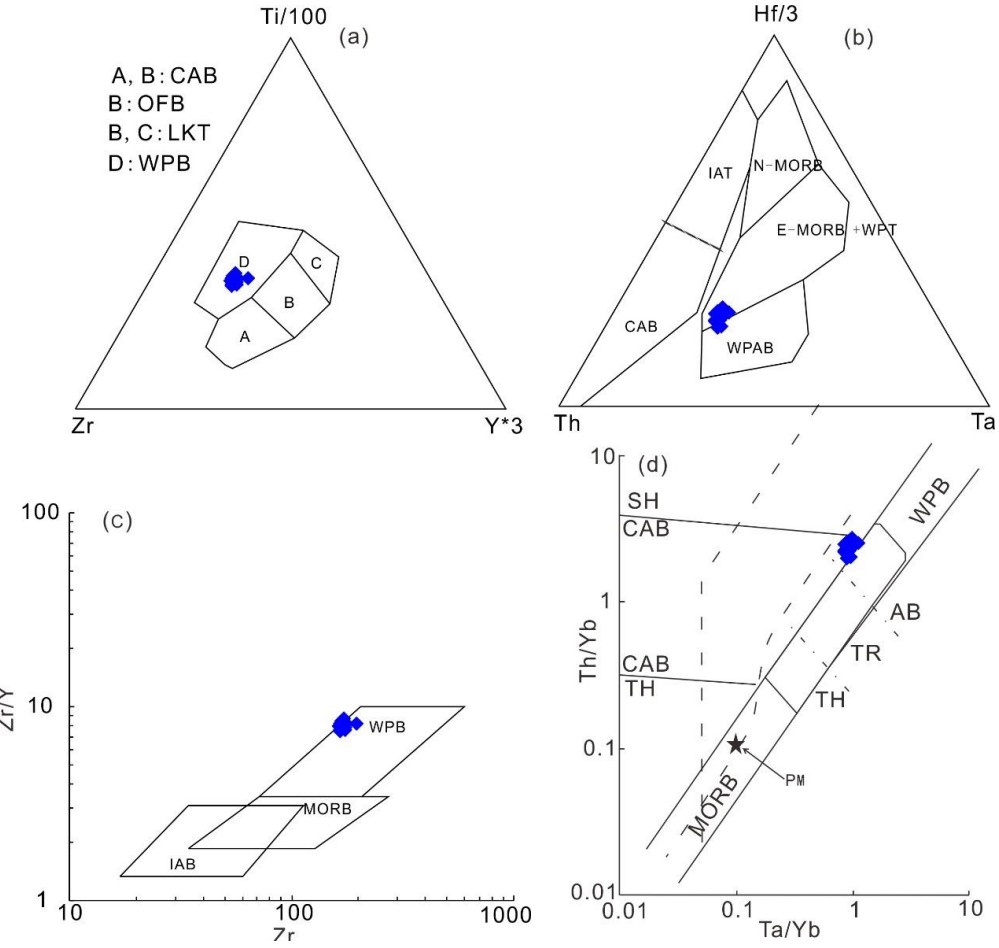

**Figure 10.** Tectonic environment discrimination diagrams for volcanic rocks from Jiangling Basin. (**a**) Diagram of Ti/1000 vs. Zr vs. Y × 3 (after [81]); (**b**)diagram of Hf/3 vs. Th vs. Ta (after [80]); (**c**) diagram of Zr/Y vs. Zr (after [81]); (**d**) diagram of Th/Yb vs. Ta/Yb (after [80]). MORB/OIB mid-ocean ridge basalt/ocean island basalt; WPB-intraplate basalts; LKT-low-K arc tholeiite; CAB-calc-alkaline basalt; SH-mugearite; TH-tholeiite; TR-transition basalt; IAB/IAT-island-arc tholeiites; and PM-primitive mantle.

*5.2. Implications for the Cenozoic Intraplate Volcanism in China*

The coastal area in Eastern China is an integral part of the western Pacific continental margin. A series of rift basins and intermediate-mafic volcanic rocks were generated via extension since the Cenozoic. Many studies (such as Sr-Nd-Pb isotope studies) of these volcanic rocks indicated that the Cenozoic volcanic rocks from Northeastern and Northern China had the source from both DMM (depleted mantle) and EMI (enriched mantle I). However, the Cenozoic volcanic rocks from Southern China and the South China Sea Basin show a tendency of DMM + EMII mixing [30,31,33–35,49,80]. The Pb isotope data from the Mesozoic and Cenozoic of Eastern China had an isotopic zonation [31,33,49,56]. The mantle in Northern China has lower $^{206}$Pb/$^{204}$Pb (<18.0) values that resemble EMI (Bau et al., 1991), whereas the mantle from Southern China has higher $^{206}$Pb/$^{204}$Pb (>18.0) values that resemble EMII [31,49,50,56,82] (Figure 9). All data plots within the field corresponded to the Cenozoic basalts of Eastern China [31,33,49,50,83]. Both basanite samples with high $\varepsilon_{Nd}$ and low$^{87}$Sr/$^{86}$Sr ratios are from the EMII mantle. There are many discussions about the location and timing of isotopic zonation because the impaction is from the Mesozoic continental collision between the North China Craton and the South China Block, as well as the post-orogenic extension in China. Recently, Cong et al. (2001) [66] found that the Tertiary volcanic rocks of a basin in the northern part of the orogenic belt had high $^{206}$Pb/$^{204}$Pb (>17.5) values, suggesting the large-scale mantle upwelling in the north part of the orogenic belt had begun in the Oligocene instead of during the Paleocene. Thus, the isotopic for the mantle in Eastern China started in the Oligocene. However, compared with the Cenozoic volcanic rocks from both sides of the Tan-Lu fault zone, it was suggested that the differences in the lithosphere between the North China Craton and the South China Block were inherited from the Mesozoic rocks feature [33,84].

The Cenozoic volcanic rocks occurred in Southern China that can be used to explore the inheritance of lithospheric mantle signatures in the south of China since the Cenozoic. The Tertiary basaltic rocks of the Jiangling Basin indicate that the Tertiary volcanic rocks reflect the lithospheric mantle according to the EMII type [85]. The properties of the Southern China Mesozoic lithospheric mantle have varying descriptions and have been described similar to EMII-type mantle [86,87]. Although other researchers have maintained that the features of the continental Mesozoic lithospheric mantle of Southern China are similar to mixed EMII mantle features [88,89], Wang (2003) [78] argues that the Chenzhou-Linwu fault is the boundary. The Yangtze block composed of Cenozoic volcanic rock has a hybrid OIB + EMII trend [85,90]. Isotopic data of $^{206}$Pb/$^{204}$Pb ratios, from the Yangtze River in the southern slope of the Dabie Mountain volcanic rocks, indicate that they are mainly associated with EMII-type lithospheric mantle [91,92]. Therefore, the Paleogene volcanic rocks from the northern margin of the Yangtze block in Jianghan Basin, which has similarities to those of EMII, might have inherited the lithospheric mantle generation properties of Mesozoic rocks. So, the volcanic rocks were generated in the tectonic setting of lithospheric extensional. The circular upwelling caused the lithosphere–asthenosphere interactions associated with soft convection currents, which resulted in the partial melting of the lithospheric mantle with EMII-type components.

## 6. Conclusions

1. Accurate $^{40}$Ar/$^{39}$Ar dating results (53.19–60.78 Ma) show that the basaltic rocks in the Jianghan Basin are formed in the Paleogene;
2. The enrichment of LREEs ((La/Yb)$_N$ = 6.14–11.72) and LILEs show that the rocks were from enriched lithospheric mantle;
3. The $^{87}$Sr/$^{86}$Sr isotopic ratios and $\varepsilon_{Nd}$ values for the Jiangling volcanic rocks range from 0.7043 to 0.7072 and +2.38 to −2.22, respectively, where the lithospheric mantle was derived from EMII and the magma was derived from partial melting of EMII-type lithospheric mantle.

**Author Contributions:** Conceptualization L.S. designed the research and prepared the original manuscript C.W. revised the manuscript K.Y. and X.Y. participated in sample collection and processing J.W. and D.L. data curation R.L. and C.Y. All authors have read and agreed to the published version of the manuscript.

**Funding:** This research was funded by the National Natural Science Foundation of China (Nos. U20A2092, 42002106, 41907262, and 41502089), the Central Welfare Basic Scientific Research Business Expenses (Nos. KK2005, KK2110, K1415, and YK1603), the China Geological Survey (Nos. DD20221684, DD20190816, DD20190817, and DD20190606), and the National Basic Research Program of China (973 Program) (No. 2011CB403007).

**Data Availability Statement:** Data are available based on request.

**Acknowledgments:** We thank Xiaolin Hao, Haibing Hu, and Peng Chen for assisting and the State Key Laboratory for Continental Tectonics and Dynamics, Institute of Geology, Chinese Academy of Geological Science and the China National Research Center for performing the geochemical analyses. Thanks to Teresa Ubide, Associate Professor, University of Queensland, Australia, for her valuable comments and suggestions, which significantly improved the quality of the paper.

**Conflicts of Interest:** The authors declare that they have no conflict of interest.

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
