# Peer review of "40Ar/39Ar Geochronology, Geochemistry and Petrogenesis of the Volcanic Rocks in the Jiangling Basin, China"

_minerals, doi:10.3390/min12091099_

Round 1

Reviewer 1 Report (Previous Reviewer 1)

After carefully going through the revised version of the manuscript, I consider that most of the comments and suggestions I raised during the first round of the review have been readily addressed. I, therefore, recommend its possible publication in the journal. Still, I suggest the authors carefully check the language and formatting (e.g., L136-144) issues during or before the proof stage, as well as use more appropriate font sizes and legend colors in e.g., Figs. 1, 6, 7, and 8.

Author Response

Response to editor and reviewer

Dear Editor,

Manuscript ID: minerals-1869785

We would like to thank Minerals for giving us the opportunity to revise our manuscript.

We also thank the reviewers for giving us constructive and thoughtful comments on previous draft. We have carefully taken their comments into consideration in preparing our revision, which has resulted in the manuscript that is clearer and more compelling. The following summarizes how we responded to reviewer’ comments.

Below is our response to comments.

Thanks for all the help.

Best wishes

Revision - authors’ response

  1. English language and style are fine/minor spell check required.

Reply: We revise the whole manuscript, and the language was checked by relevant experts. Thanks to Teresa Ubide, associate professor, University of Queensland, Australia, for her valuable comments and suggestions, which significantly improved the quality of the paper.

  1. After carefully going through the revised version of the manuscript, I consider that most of the comments and suggestions I raised during the first round of the review have been readily addressed. I, therefore, recommend its possible publication in the journal. Still, I suggest the authors carefully check the language and formatting (e.g., L136-144) issues during or before the proof stage, as well as use more appropriate font sizes and legend colors in e.g., Figs. 1, 6, 7, and 8.

Reply: We have checked the language and formatting (e.g., L136-144) issues before the proof stage, as well as use more appropriate font sizes in e.g., Fig. 1. The meaning of different legends colors in the relevant figures have been clarified and detailed. Fig. 6 just wanted to show the distribution REE patterns, which has achieved the goal, so we did not add the sample number. This paper has done a lot of field and experimental work. We hope that the reviewer can give your approval.

Reviewer 2 Report (Previous Reviewer 2)

The manuscript is now improved. Yet, English should be re-examined and corrected.

Author Response

Response to editor and reviewer

Dear Editor,

Manuscript ID: minerals-1869785

We would like to thank Minerals for giving us the opportunity to revise our manuscript.

We also thank the reviewers for giving us constructive and thoughtful comments on previous draft. We have carefully taken their comments into consideration in preparing our revision, which has resulted in the manuscript that is clearer and more compelling. The following summarizes how we responded to reviewer’ comments.

Below is our response to comments.

Thanks for all the help.

Best wishes

Revision - authors’ response

  1. Moderate English changes required.

Reply: Thanks to Teresa Ubide, associate professor, University of Queensland, Australia, for her valuable comments and suggestions, which significantly improved the quality of the paper. We revise the whole manuscript, and the language was checked by relevant experts. Earlier Manuscript of the article had also been polished by professional bodies.

This manuscript is a resubmission of an earlier submission. The following is a list of the peer review reports and author responses from that submission.

Round 1

Reviewer 1 Report

I appreciate this opportunity to review the manuscript entitled ‘40Ar/39Ar geochronology, geochemistry and petrogenesis of the volcanic rocks in the Jiangling Basin, China’ by Wang et al. After carefully going through the paper, I consider that there leaves a large room for its further improvement and a moderate revision is required. My main comments are presented below for potential consideration.

The Introduction can be improved by further emphasizing the progress and potential issues of previous relevant studies and the scientific necessity of this study. The description made by the authors in L56-59 is also unjustified.

The section of Geological setting and the geological map are too simplified. A further expansion and revision of them are required because the authors apparently intend to probe the significance of the studied volcanics beyond the scale of Jiangling Basin. Some parts of the Geological Setting are provided in the Discussion e.g., L280-285.

Similar to the above comments, because the authors intend to make comparisons of geochronology and composition across basins or regions, it would be greatly beneficial to directly show them in the figures. In this regard, many figures of geochemical diagrams need major improvements. Please also make sure the key geological elements (e.g., Jiangling Basin, South China, and the South China Sea) are clearly marked in relevant figures such that readers could easily locate them.

For the sections of samples, methods, and results, I did not see the detailed sampling info (either in the table or figure) and analytical methods of whole-rock isotopes.

Some figures need minor revisions in clarifying the meaning of different legends (e.g., different colors used in Figs. 6 and 7) as well as expanding captions (e.g., Figs. 1, 3, and 8). By the way, the order of Figs. 9 and 10 in the text seems wrong.

The language of the manuscript is generally ok. However, small grammatical errors, repetitions, and inappropriate expressions are not uncommon in the full text. Please carefully check the language before resubmission and may ask for the professional help from native speakers. Please insert key citations where necessary in the text (e.g., L262) and manage to keep them as concise as possible (e.g., L47-51). Give full names before abbreviations (e.g., L240, 183). Besides, please note to present the sentences, especially those in the Discussion, in a more specific way such that to avoid confusion (e.g., L271, 300).

Reviewer 2 Report

minerals-1791247 comments.

The manuscript is dealing with an interesting topic, the boundary between mantle end-members underlying Northern and Southern China blocks.

Unfortunately, it seems as if the authors have changed their mind in the discussion chapter, starting from EMI mantle end-member, like below Northern China, and preferring in the last page EMII mantle end-member, like below Southern China, without any clear explanation. A map showing where these mantle end-members occur below Eastern China would be useful for the reader not fully aware of the EMI-EMII conundrum.

The Ar-Ar method is not the best way to determine the eruption ages, because rocks are altered and it is not clear what kind of materials was selected for isotopic analysis.

The English is poor and should be checked by a person well aware of English grammar and pitfalls.

Line 87. What are the large light crystals in figure 3c?

Lines 108-109.The sentence is incomplete. Amphibole and jadeite were not described in the petrographical subchapter. What species do amphibole crystals correspond to? Where do jadeite crystals come from: metamorphic xenocrysts?

Table 1. Please explain why (H2O+ + CO2) > LOI (for example, in BO1 sample, H2O+ = 1.1% and LOI = 0.51%).

Line 129. Use of the TAS diagram is compulsory. After recalculation on an anhydrous basis (see Le Maitre et al., 2002), all samples correspond to basaltic andesite (recalculated SiO2 contents higher than 52%) and plot near the boundary with basaltic trachyandesite.

Figure 4b. It is doubtful that the K2O-SiO2 diagram comes from Winchester and Floyd (1977).

Lines 166-217. The Ar-Ar method is prone to systematic errors in samples having suffered low-temperature (chlorite, prehnite) alteration. It is not clear what was analysed and the mention of amphibole and jadeite (line 108) is puzzling. The results are scattered and may not represent eruption ages.

Lines 194-195. "Did not include" is not correct. The correct words are "were not included".

Lines 205-206. What do you mean by "a plateau age of 1420°C"?

Line 240. The TAS diagram is missing (see comment on line 129).

Line 258. Please be coherent. Do Jiangling Basin samples correspond to subalkaline, or to alkali basalt groups? Geochemical figures 4 and 9 rather favour alkali basalt group.

Lines 280-331. This chapter should be re-organized and re-written. The English is poor, with too many incomplete sentences. The text is too wordy and should be reduced. The EMI-EMII conundrum is poorly discussed, as many arguments are misleading. After long considerations on EMI features displayed by the samples, the EMII end-member is claimed as the source of the rocks. The reviewer is completely lost and does not understand what is going on.

Line 282. Instead of "have generated", please write: "were generated".

Lines 288-290, 297-299. These sentences are not understandable. Please rephrase.

Line 316. This is not understandable. Do Jiangling Basin rocks display EMI features, as claimed in the previous pages, or EMII features, as written here?

To sum up, the manuscript is far from being ready for publication. It should be re-thought and re-written.